# Illuminating the effect of beneficial blue light and ROS-modulating enzymes in Dupuytren's disease

Carina Jaekel, Simon Thelen*, Lisa Oezel, Marie H. Wohltmann, Julia Wille, Joachim Windolf, Vera Grotheer

Department of Orthopedics and Trauma Surgery, Medical Faculty, Heinrich Heine University, Düsseldorf, Germany

* simon.thelen@med.uni-duesseldorf.de

**Data Availability Statement:** All relevant data are within the manuscript and its Supporting information files.

## Abstract

Dupuytren's disease (DD) is a fibroproliferative disorder of the palmar aponeurosis, which is characterized by a compound myofibrogenesis and evidenced by an increased expression of α-smooth muscle actin (α-SMA). In Dupuytren's tissue, higher levels of reactive oxygen species (ROS) are documented, stimulating the proliferation and differentiation of myofibroblasts. Our preliminary study demonstrates that α-SMA-expression is significantly inhibited by blue light irradiation in DD. The objective of this study was to investigate the beneficial effect of blue light irradiation and to elucidate the influence of ROS on myofibrogenesis in the pathogenesis of DD. Therefore, an in-vitro model of human DD fibroblasts was used. DD fibroblasts and control fibroblasts isolated from carpal tunnel syndrome (CTS) were daily irradiated with 40 J/cm$^2$ ($\lambda$ = 453 nm, 38 mW/cm$^2$). Protein expression of ROS-modulating enzymes (Catalase, NOX4, SOD1, MnSOD) and α-SMA were determined, and additionally analysed after a pharmacological inhibition of the TGF-β1-signaling with SB431542. Furthermore, the protein expression of α-SMA as surrogate parameter for myofibrogenesis was evaluated after applying different concentrations of long-lasting ROS. It could be determined that the beneficial blue light irradiation, which inhibited myofibrogenesis, is mediated by a significant inhibition of catalase protein expression. This effect should be accompanied with an increased intracellular ROS level. Proof of evidence was an H$_2$O$_2$-application on DD fibroblasts, also leading to a decreased myofibrogenesis. Furthermore, it could be demonstrated that endogenous MnSOD was significantly downregulated in resting DD fibroblasts. If DD fibroblasts were treated with the pharmacological inhibitor SB431542, myofibrogenesis was inhibited, but MnSOD expression was simultaneously elevated, which ought to affect ROS level by raising intracellular H$_2$O$_2$ amount. Blue light irradiation as well as the pharmacological action of SB431542 in consequence mediates their beneficial effect on disturbed myofibrogenesis in DD by further increasing ROS level. The present study demonstrates the importance of intracellular ROS homeostasis in DD and illuminates the beneficial effect of blue light as a promising therapy option for DD.

**Funding:** This research did not receive any specific grant from funding agencies in the public, commercial, or not-for-profit.

**Competing interests:** All authors certify that they have no affiliations with or involvement in any organization or entity with any financial interest (such as honoraria; educational grants; participation in speakers' bureaus; membership, employment, consultancies, stock ownership, or other equity interest; and expert testimony or patent-licensing arrangements), or non-financial interest (such as personal or professional relationships, affiliations, knowledge or beliefs) in the subject matter or materials discussed in this manuscript.

## Introduction

Dupuytren's disease (DD) is a fibroproliferative disorder of the palmar fascia of the hand that often leads to disabling flexion contracture of the affected fingers. The prevalence of DD increases with age with a male to female ratio of 7:1 and a range from 2.4% to 42% as well as higher prevalence in northern European countries [1–3]. Its multifactorial etiology is a complex interaction between genetic predisposition, aging, environmental factors and deranged protein expression [4, 5]. The pathological process of DD is described as a disturbed and uncontrolled myofibrogenesis, leading to the development of nodules and to the formation of characteristic cords [6]. Myofibroblasts are cells containing features of fibroblasts and smooth-muscle-cells, as they are characterized by a cytoplasmic microfilament protein called α-smooth muscle actin (α-SMA) [7, 8]. Preliminary studies confirmed that transforming growth factor-β 1 (TGF-β1), a potent pro-fibrogenic cytokine abundant in DD, induces myofibroblast proliferation and differentiation with significant higher α-SMA expression [9]. It is known that TGF-β1 increases ROS production and suppresses antioxidant enzymes, thereby inducing redox imbalance, which contributes to fibrosis [10]. Accordingly, it could be demonstrated that high levels of reactive oxygen species (ROS) are generated in Dupuytren's tissue [9]. Moderate concentrations of ROS are known to be important to regulate cellular differentiation, proliferation and migration by several signaling pathways. A redox imbalance in favor of ROS, called oxidative stress, causes undirected cell damage through oxidation of macromolecules, such as proteins or DNA and RNA, frequently inducing deleterious damage and apoptosis in cells [11–13]. Therefore, oxidative stress is related in the pathogenesis of many diseases like cancer, diabetes mellitus, atherosclerosis, as well as in aging and tissue fibrosis [14–16]. The involvement of ROS in the pathogenesis of DD could possibly explain some epidemiological associations with increasing age, alcohol or cigarette abuse, diabetes and human immunodeficiency virus (HIV) [17, 18]. Intracellular ROS concentrations under physiological conditions are balanced in between their rates of production and protective antioxidant cellular mechanism. ROS homeostasis is mediated by different enzymes as NADPH Oxidase (NOX) and Xanthine Oxidase (XO), producing superoxide ($O_2^-$), the most reactive oxygen radical [19]. Superoxide Dismutase 1 (SOD1) as well as Glutathione Peroxidase (GPX), which can catalyze $O_2^-$ to hydroperoxide ($H_2O_2$), a less reactive ROS. Catalase in turn promotes the conversion from $H_2O_2$ into $H_2O$.

Our previous *in vitro* study demonstrates that initial elevated and activated α-SMA expression in DD fibroblasts can be significantly inhibited by blue light irradiation (λ = 453 nm, 40 J/$cm^2$, 38 mW/$cm^2$) [20]. With a longer the blue light treatment, α-SMA protein expression decreases. Considering this fact, blue light application could be an auspicious treatment option for DD. But the mode of action based on blue light therapy among DD fibroblast has not been explored until now. Taflinski *et al.* assumed that blue light irradiation with the more increased/higher energetic wavelength of 420 nm led to an increased amount of ROS as well as an α-SMA reduction in dermal fibroblasts. It was supposed, that the induction of ROS was responsible for the α-SMA inhibition [21]. This finding was rather interesting, because as described in aforementioned DD studies, the already existing ROS burden was associated with the pathogenesis of DD [14, 15]. Our previous study could also determine an elevated ROS amount in DD fibroblasts and that blue light treatment with a wavelength of (λ = 453 nm), further gains ROS formation [20]. This raises the question of how blue light irradiation affects the intracellular ROS level and controls DD pathology and thus its therapy.

The aim of this study was to evaluate the interaction between the application of blue light and intracellular ROS level in DD. Thereby, the following questions should be answered:

(1) Which mechanism contributes the blue light mediated inhibition of myofibrogenesis? and
(2) What functions do ROS assume in DD myofibrogenesis and pathology?

## Materials and methods

### Patients

The present study was approved by the local Research Ethics Committee of the Heinrich-Heine-University Duesseldorf (no. 5882R). After giving written consent, the patients underwent either palmar fasciectomy or carpal tunnel release at the Department of Orthopedics and Trauma Surgery, University Hospital, Düsseldorf, Germany. Patients who suffered from Dupuytren's disease (DD) served as patient group. Patients suffering from carpal tunnel syndrome (CTS) with normal palmar fascia served as control group. During the operation, tissue samples were deposited in PBS (1% penicillin/streptomycin) and afterwards cooled to 4°C. Samples were processed within the first 24 h. All data were pseudonymised prior to analysis. The use of human tissue was in compliance with the Declaration of Helsinki. Nineteen patients suffering from DD (female: 1, male: 18, mean age: 63 years) and seventeen patients suffering from CTS (female: 10, male: 7, mean age: 60 years) were included in the study.

### Materials

Chemicals were obtained from Sigma-Aldrich, and cell culture materials from Cellstar (Greiner bio-one), if not mentioned otherwise.

### Human fibroblasts isolation and cultivation

Fibroblasts were isolated from Dupuytren fibrous tissue, respectively CTS tissue (cleared from fat and blood). Briefly, tissue samples were washed twice with phosphate buffered saline (PBS), cut into small pieces (0.2 x 0.2 mm) applied in a petri dish (10 cm) with 13–15 ml DMEM ((4.5 g/l glucose, with α-glutamin (ThermoFisher, Massachusetts, USA)) with 10% FBS, 1% penicillin/streptomycin, 1 x sodium pyruvate (as in the following described as standard medium) and incubated at 37°C by 5% $CO_2$. Outgrowing fibroblasts were monitored microscopically daily and the medium was changed twice a week. Upon reaching confluence (ca. 70%–80%), the fibroblasts were characterized, further cultured, seeded or cryopreserved. Confluency of the DD fibroblasts was reached, on average, 14 days after extraction. The experiments could begin after a further 14 days, usually in cell culture passage 3–5, in exceptional cases up to cell culture passage 8. The prerequisite for using cell culture passage 8 is that fibroblasts proliferation rate and morphology had not changed. CTS or DD fibroblasts were washed with PBS and detached after incubation with 1% trypsin/EDTA solution at 37°C. Detachment was continuously monitored with light microscopy and neutralized with the corresponding amount of cell culture media. Characterization was performed by immunocytochemistry staining using a mouse anti-human vimentin antibody according to the manufacturer's instructions (Invitrogen, Darmstadt, Germany; VA2922991 (1:200)) prior to analysis. All experiments were performed in cell culture passages 3–8. Twenty-four hours before experiments were started fibroblasts were seeded in a density of 3 x $10^4$ cells/6-Well.

### Blue light irradiation

The blue light application was accomplished with a prototype of a narrow band light-emitting diode (LED), which emits monochromatic light of the wavelength λ = 453 nm +/- 10 nm. The light emitting device used in our experiments was produced by Philips Technology Research Laboratories, (Aachen, Germany). Sixty (6 x 10) LEDs were equally distributed over a total of

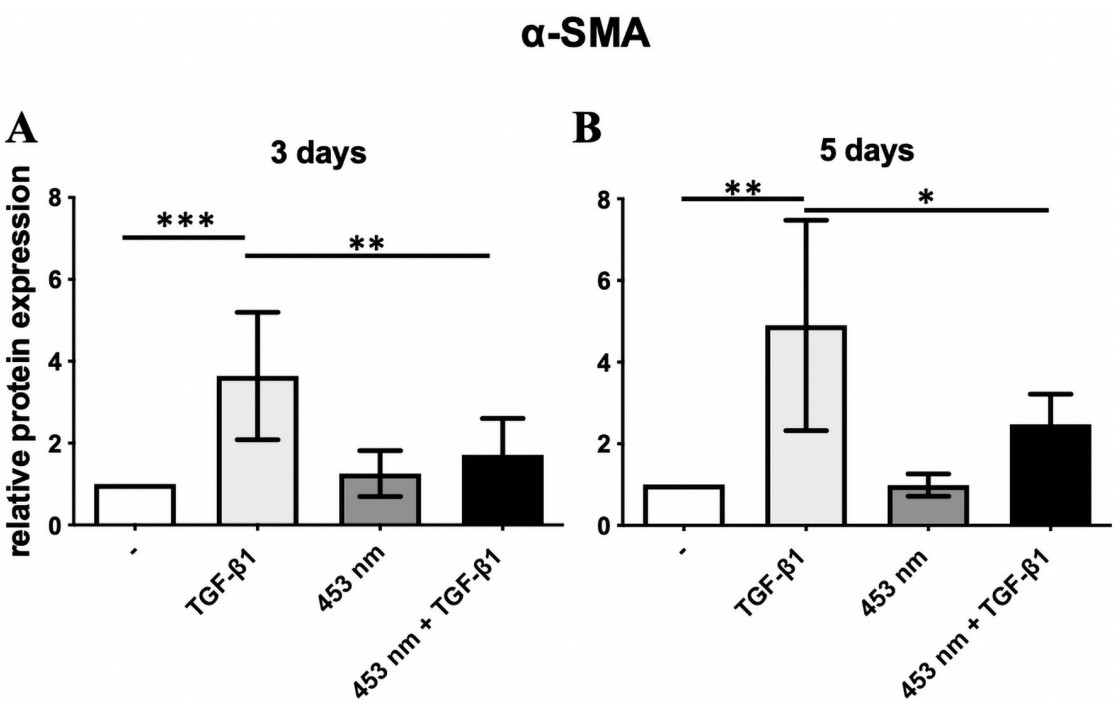

**Fig 1. Relative α-SMA protein expression of irradiated DD fibroblasts (40 Joule/cm$^2$) activated with TGF-β1.** α-SMA protein expression in DD fibroblasts after 3 (A) and 5 (B) days. TGF-β1 increased α-SMA protein expression on day 3 (A) and 5 (B). Blue light irradiation reduced α-SMA protein expression significantly in DD fibroblasts activated with TGF-β1 on day 3 (A) and 5 (B). * p ≤ 0,005; ** p ≤ 0,005; *** p ≤ 0.0005. Bars represent mean ± SD of individual experiments indicated (n = 8).

10 x 12 cm. Before being released for delivery the irradiance of the LED-device was validated by the manufacturer using an integrating (Ulbricht) sphere. The power density was 38 mW/cm$^2$, at a distance of 8 cm, the time of exposure was 16.7 min, according to 40 J/cm$^2$, a non-toxic dose [20]. The exact irradiation time was evaluated using the precise hand radiometer RM 21 (Dr. Gröbel UV-Elektronik GmbH (Ettlingen, Germany). Fibroblasts were continuously irradiated from above in PBS in transparent 6-Well-plates. For the short irradiation time (16.7 min) those samples of CTS and DD fibroblasts were exposed at room temperature (18–21˚C) under the LED device. For the same duration time control samples of DD and CTS fibroblasts were stored in a *Biometra OV3 Hybridisation* oven at 25˚C. The aim was to have comparable temperatures in both experimental setups (+/- irradiation). This was evaluated in control experiments using a digital thermometer, ensuring that irradiated DD and CTS fibroblasts never exceeded a temperature of 38˚C while control fibroblasts were kept at a temperature of 37˚C. After irradiation, PBS was replaced by fresh media. To evaluate the influence of blue light application, DD as well as CTS fibroblasts were treated for 3 and 5 days. Fibroblasts were daily activated with (+/-) 1 ng/ml TGF-β1 (PeproTech GmbH, Hamburg, Germany), or irradiated (40 J/cm$^2$), or treated and irradiated, or neither of those (Fig 1).

## H$_2$O$_2$ treatment

To investigate the effect of H$_2$O$_2$, we evaluated the α-SMA protein expression by western blot analysis (see below) after a daily application of TGF-β1 and H$_2$O$_2$ in different non-toxic concentrations (1/5/10 ng/ml TGF-β1; 10/20/50 μM H$_2$O$_2$) and an incubation period of 5 days. Previously conducted studies by our working group determined non-toxicity of TGF-β1 and H$_2$O$_2$ using a CellTiter-Blue Cell Viability Assay from Promega. According to the results, we

conclude that the use of TGF-β1 (1 ng, 5 ng, 10 ng/ml) and $H_2O_2$ (10 μM, 20, μM, 50 μM) over the course of 5 days has no toxic effect (S1 Fig).

## Glutathion Peroxidase assay

Glutathione Peroxidase (GPX) activity was measured by using a GPX Assay Kit, Abnova. For sample preparation, fibroblasts were homogenized in 100 μl PBS. After sonification and centrifugation (15 min at 3,000 g; 4˚C) supernatant was collected. According to the manufacturer's protocol 12.5 μl or 25 μl supernatant were given in a 96-well plate and adjusted to a final volume of 50 μl with Assay Buffer. At first 12.5 μl supernatant were used and if the achieved mathematical value was not within the linear range, the experiment was repeated with 50 μl supernatant. Thereafter, 40 μl of the reaction mix (33 μl assay buffer, 3 μl, 40 mM NADPH solution, 2 μl GR solution) were given to the samples. 5–10 μl GPX positive control was used as positive control (PC), assay buffer was used as a reagent control (RC). A NADPH standard curve was performed. After an incubation of 15 min GPX reaction was started by adding 10 μl cumene hydroperoxide solution. The measurement of GPX activity is accomplished via the consumption of NADPH, via measuring the absorption at 340 nm. (ΔA340 nm = [(Sample_A1-Sample_A2)–(RC_A1 –RC_A2)]; A1 = 0 min, A2 = 5 min.).

## Western Blot analysis

A Western Blot analysis was performed to determine the protein expression of α-SMA, on the one hand, and ROS-modulating enzymes (Catalase, NOX4, SOD, MnSOD) as well as their transcription factors (NFκB, β-catenin) on the other. Protein determination was performed with the Pierce BCA Protein assay Kit (ThermoFisher). In general, 10 μg of protein or in the case of NOX4, Mn-SOD or NFκB 20 μg of protein were mixed with 5 μl Laemmli buffer (4 × Trisglycin-SDS sample buffer, 252 mmol TrisHCL pH 6.8; 40% Glycerin; 8% SDS; 0.01% bromphenol blue + 20% mercaptoethanol), centrifuged (4,000 g, 5 min at 4 ˚C) denaturated for 5 min at 95˚C, and separated on a 12% sodium dodecyl sulphate-polyacryl-amide gel (SDS-PAGE). Separated proteins were transferred to a nitrocellulose membrane (BioRad Trans-Blot Turbo). Following, the membranes were saturated with the different antibodies (Table 1). The antibodies were incubated at 4˚C overnight. Anti-rabbit or anti-mouse conjugated with horseradish peroxidase (HRP) served (1:1000) as a second antibody with 0.025% anti-Western marker in TBST, that was added for 1 h (RT). Before and after the addition of

**Table 1. Antibodies.**

| Antibody | Manufacturer | Catalogue No | Species | Application |
|---|---|---|---|---|
| α-smooth muscle actin | Abcam | Ab7817 | Mouse | 1:1000 in 5% BSA |
| β-Catenin | Abcam | Ab16051 | Rabbit | 1:3000 in 5% BSA |
| Mn-SOD | StressMarq Biosciences | SPC-118C/D | Rabbit | 1:4000 in 5% BSA |
| Katalase | Origene | TA502564 | Mouse | 1:3000 in 5% BSA |
| SOD1 | Cell Signaling Technology | 2770 | Rabbit | 1:1000 in 5% BSA |
| SOD1 | StressMarq Biosciences | SPC-116-D | Rabbit | 1:1000 in 5% BSA |
| NFκB | Santa Cruz Biotechnology | Sc-8414 | Mouse | 1:500 in 5% Milch |
| NOX4 | Novus–a biotechne brand | NB110-58849 | Rabbit | 1:1000 in 5% BSA |
| **Antibody** | **Manufacturer** | **Catalogue No** | **Application** | |
| Polyclonal Goat anti-Mouse Immunoglobulin/HRP | Dako | P0447 | 0,1% | |
| Polyclonal Goat anti-Rabbit Immunoglobulin/HRP | Dako | P0448 | 0,1% | |

the second antibody, the membranes were washed three times with TBS-T. Western blots were analyzed with image lab version 6.0.1 build 34, 2017, Standard Edition, BioRad laboratories.

### The inhibition of transforming growth factor (TGF-β1)-signaling with SB431542

In order to investigate the influence of the inhibitor SB431542, fibroblasts were sown and treated for 5 days. The medium was aspirated daily and the cells were washed with 1 ml PBS. 1 ml of standard medium (cf. page 5) was added to the control cells and 1 ml of medium with 0.5 μmol SB 431542, an ALK receptor 4, 5 and 7 inhibitor, were added to the cells in the treatment group. Afterwards 0, 1, 5 or 10 μg TGF-β1 were given to the SB-treated and untreated fibroblasts. Finally, the expression of α-SMA, Catalase, NOX4 and MnSOD protein expression was examined.

### Statistical analysis

Statistical analysis was performed using GraphPad Prism8 (GraphPad Software, San Diego, CA, USA). Real-valued data was first tested for normal Gaussian distributions with Kolmogorow Smirnow test. Statistical analysis was realized with one-way or two-way ANOVA and Friedman-test, followed by post-hoc Bonferroni test. The data were expressed as mean value and standard deviation (SD). $P \leq 0.05$ and below were considered significant.

## Results

### Irradiation-based inhibition of α-SMA protein expression

TGF-β1 increased α-SMA protein expression on day 3 and 5 in non-irradiated DD fibroblasts. Blue light irradiation significantly reduced α-SMA protein expression in TGF-β1-activated DD fibroblasts on day 3 ($p \leq 0,005$) and 5 ($p \leq 0,005$, Fig 1A and 1B). Data obtained from respective CTS figures were only shown or discussed, if the CTS data were significantly different to DD data.

### Irradiation-based adjustment of ROS-modulating enzymes

Blue light application did not modify GPX activity (Fig 2A and 2B). There was only one slight effect on TGF-β1 activated CTS fibroblasts after blue light application on day 5 in comparison to untreated CTS fibroblasts. Also, the expression of SOD1 was not modified in DD fibroblasts after a blue light application on day 3 and 5 (Fig 2C and 2D). Blue light irradiation did not affect NOX4 protein expression in DD fibroblast (Fig 2E and 2F), although there was a slight effect on day 3 in TGF-β1 treated DD fibroblasts (Fig 2E).

The expression of catalase in activated DD fibroblasts (with TGF-β1) was significantly decreased compared to resting DD fibroblasts ($p \leq 0.05$). And the additional blue light application further inhibited significantly catalase expression on day 3 ($p \leq 0.0005$, Fig 3A). On day 5 catalase expression was significantly declined in activated and irradiated fibroblasts compared to activated DD fibroblasts ($p \leq 0,005$, Fig 3B).

### Effect of $H_2O_2$ treatment on α-SMA protein expression

To determine the impact of ROS on myofibrogenesis in DD, α-SMA protein expression as surrogate parameter was evaluated. To determine the impact of $H_2O_2$ on α-SMA protein expression in DD fibroblasts, different concentrations of TGF-β1 and $H_2O_2$ were applied. In conclusion, dependent on the TGF-β1 concentration (1/5/10 ng/ml TGF-β1), the α-SMA protein expression could be reduced by elevating non-toxic $H_2O_2$ amounts (Fig 4A–4C).

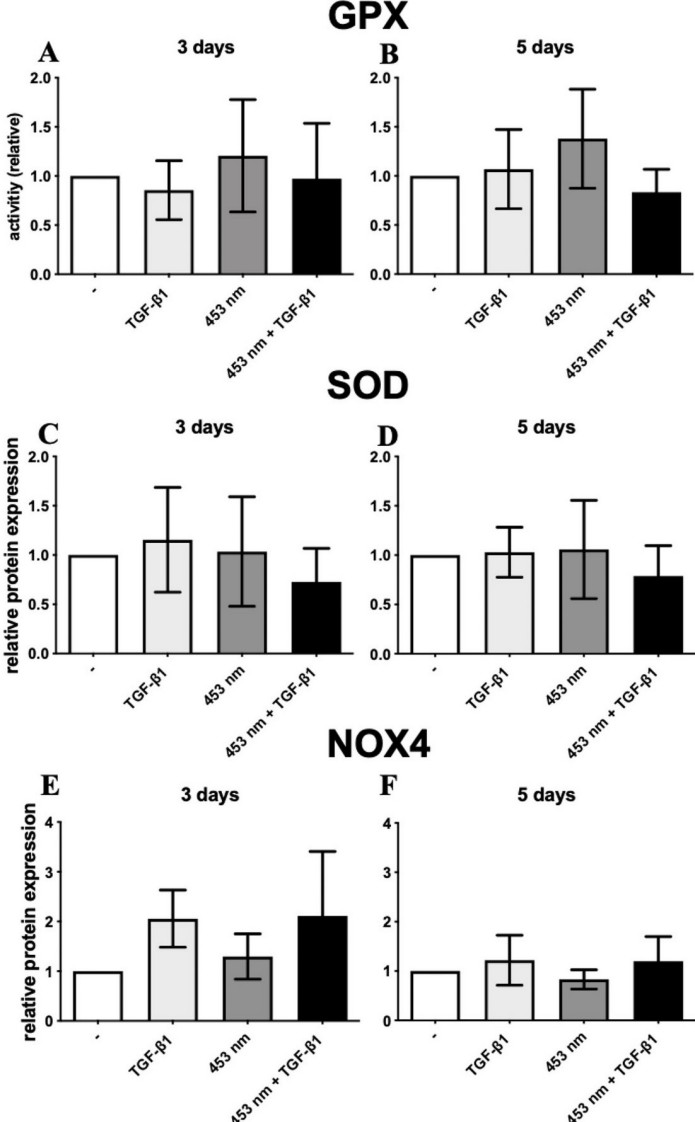

**Fig 2. GPX activity DD fibroblasts (A and B).** There is no significant difference of GPX activity on day 3 (A) and 5 (B). Bars represent mean ± SD of individual experiments indicated (n = 10). **Relative protein expression of SOD1 in DD fibroblasts (C and D).** Blue light irradiation caused no significant effect on DD fibroblasts on day 3 and 5. * p ≤ 0.05. Bars represent mean ± SD of individual experiments indicated (n = 8). **Relative protein expression of NOX4 in DD fibroblasts (E and F).** TGF-β1 caused an elevated NOX4 expression in tendency, and the blue light irradiation further increased this effect on day 3, even though this effect was not significant. On day 5 the effect was reduced. Bars represent mean ± SD of individual experiments indicated (n = 8).

## MnSOD expression in DD fibroblasts compared to CTS fibroblasts

MnSOD protein expression was significantly inhibited in resting DD fibroblasts compared to CTS fibroblasts (p ≤ 0.05). Furthermore, MnSOD expression was significantly decreased in irradiated (p ≤ 0.0005) and as well in irradiated and TGF-β1activated DD fibroblasts (p ≤ 0.05) on day 3 (Fig 5A).

On day 5, in irradiated DD fibroblasts MnSOD expression was significantly diminished compared to CTS fibroblasts (p ≤ 0.0005, Fig 5B). Moreover, in all DD fibroblasts MnSOD was in tendency reduced compared to CTS fibroblasts.

## Catalase

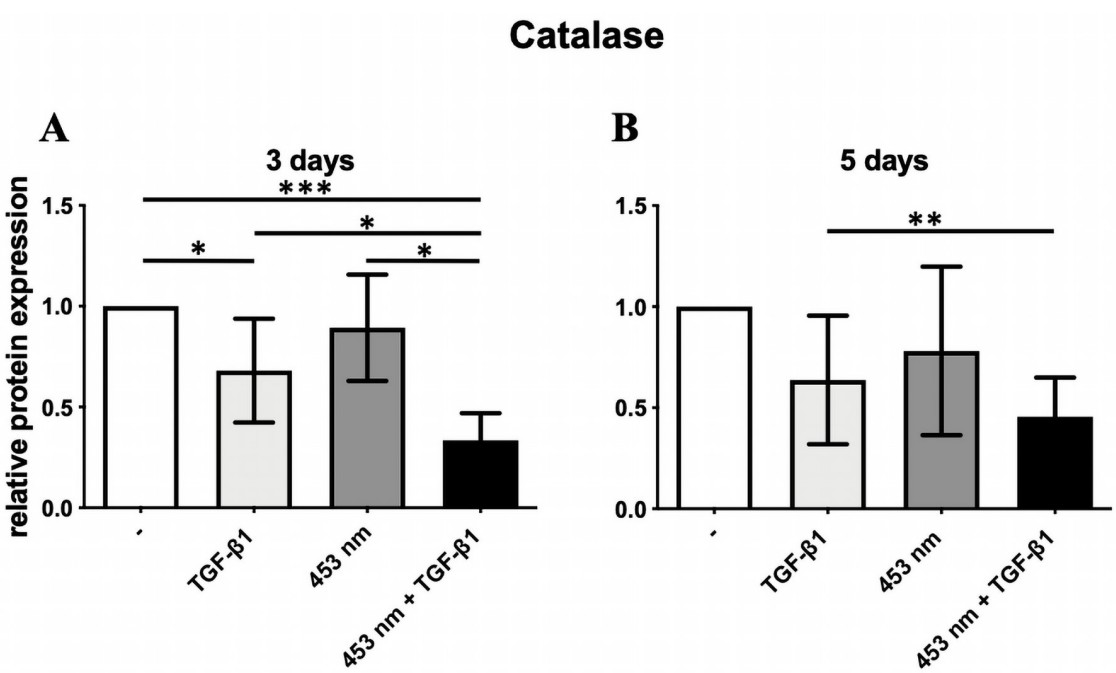

**Fig 3. Catalase protein expression in DD fibroblasts.** The expression of catalase in DD fibroblasts activated with TGF-β1 was significantly decreased on day 3(A). Blue light irradiation significantly decreased relative catalase expression in DD fibroblasts activated with TGF-β1 on day 3 and 5 (A, B). * p ≤ 0.05; ** p ≤ 0,005; *** p ≤ 0.0005. Bars represent mean ± SD of individual experiments indicated (n = 8).

### Application of SB431542 on DD fibroblasts

The selective TGF-β1 inhibitor SB431542 reduced α-SMA protein expression significantly, regardless of the TGF-β1 concentrations (p ≤ 0.0005, Fig 6A). Comparatively, the protein expression of MnSOD was significantly increased by SB431542 in DD fibroblasts (p ≤ 0.0005, Fig 6B). SB 431542 had no effect on catalase and NOX4 independent on TGF-β1 concentration (Fig 6C and 6D).

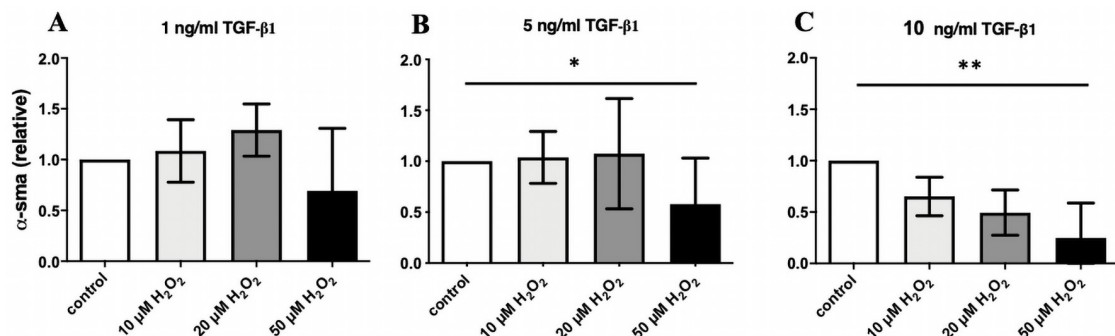

**Fig 4. α-SMA protein expression in DD fibroblasts dependent on $H_2O_2$ concentrations.** DD fibroblasts were activated with increasing concentrations of TGF TGF-β1: 1 ng/ml (A), 5 ng/ml TGF-β1 (B) and 10 ng/ml TGF-β1 (C). Dependent on TGF-β1 concentrations, α-SMA protein expression could be significantly inhibited by a rising non-toxic $H_2O_2$ amounts. * p ≤ 0.05; ** p ≤ 0,005. Bars represent mean ± SD of individual experiments indicated (n = 3).

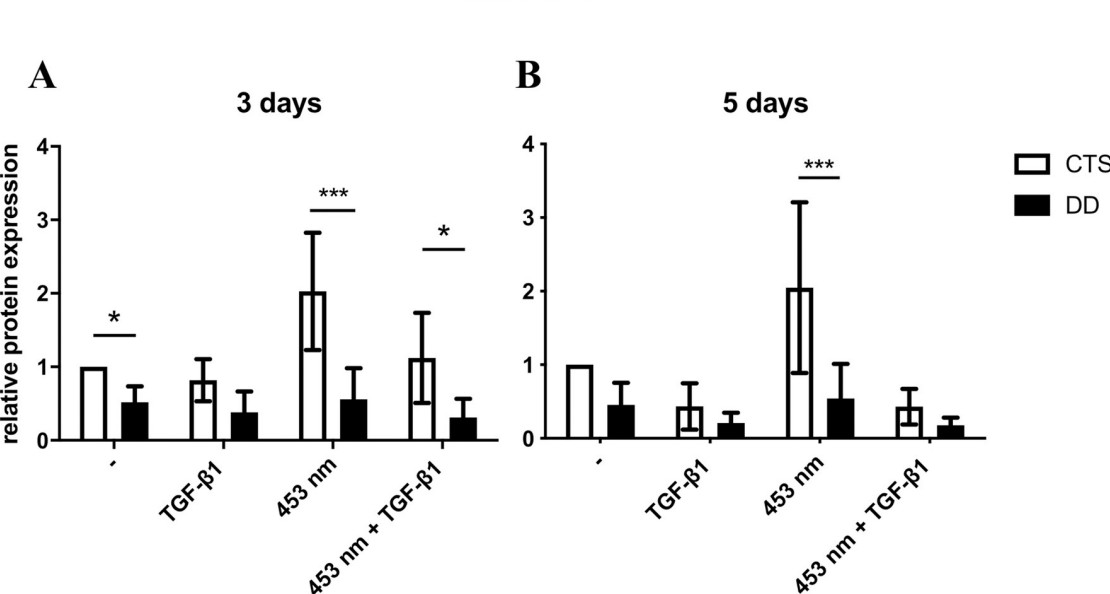

**Fig 5. MnSOD protein expression in CTS and DD fibroblasts.** The expression of MnSOD in blue light irradiated DD fibroblasts was significantly inhibited in comparison to irradiated CTS fibroblasts on day 3 (A). Furthermore, MnSOD expression was significantly decreased in irradiated and as well as in irradiated and TGF-β1activated DD fibroblasts on day 3 (A). On day 5, in irradiated DD fibroblasts MnSOD expression was significantly diminished compared to CTS fibroblasts (B). * $p \leq 0.05$; *** $p \leq 0.0005$. Bars represent mean ± SD of individual experiments indicated (n = 8).

### Irradiation-based modulation of transcription factors

**NFκB protein expression.** In irradiated and TGF-β1 activated CTS fibroblasts, NFκB protein expression was significantly inhibited in comparison to irradiated CTS fibroblasts on day 3 (p ≤ 0.05, S2A and S2B Fig). Moreover, the irradiation elevated in tendency NFκB protein expression compared to resting fibroblasts, and this effect was significant in DD fibroblasts on day 5 (p ≤ 0.05, S2A–S2C Fig).

**β-catenin protein expression.** In irradiated and activated CTS fibroblasts, the β-catenin expression was significantly reduced compared to irradiated CTS fibroblasts (p ≤ 0.05, S2D and S2E Fig) on day 3. On day 5, β-catenin expression was significantly increased in activated as well as in irradiated and activated DD fibroblasts (p ≤ 0.05, S2F Fig).

## Discussion

The main objective of DD therapy is to improve hand and digit function [22]. Different treatment options have been proposed according to the severity of the disease, underlying patient risk factors as well as the surgeon's preference: Partial fasciectomy, needle fasciotomy and enzymatic fasciotomy (injection of collagenase into the cord) are the most recent therapeutic strategies [23]. Due to the invasiveness of these treatment options and the high rate of recurrence, new and alternative therapies are desirable. Our previous studies demonstrated that the application of blue light could inhibit the differentiation of DD fibroblasts into myofibroblasts and the accompanied α-SMA protein expression [20]. This could be an auspicious treatment option for DD. Taflinski *et al.* postulated in their study with dermal fibroblasts and an irradiation with the wavelength of 420 nm, that the effect of α-SMA inhibition was probably mediated via low-level of oxidative stress [21]. In the context of DD, this assumption was noteworthy,

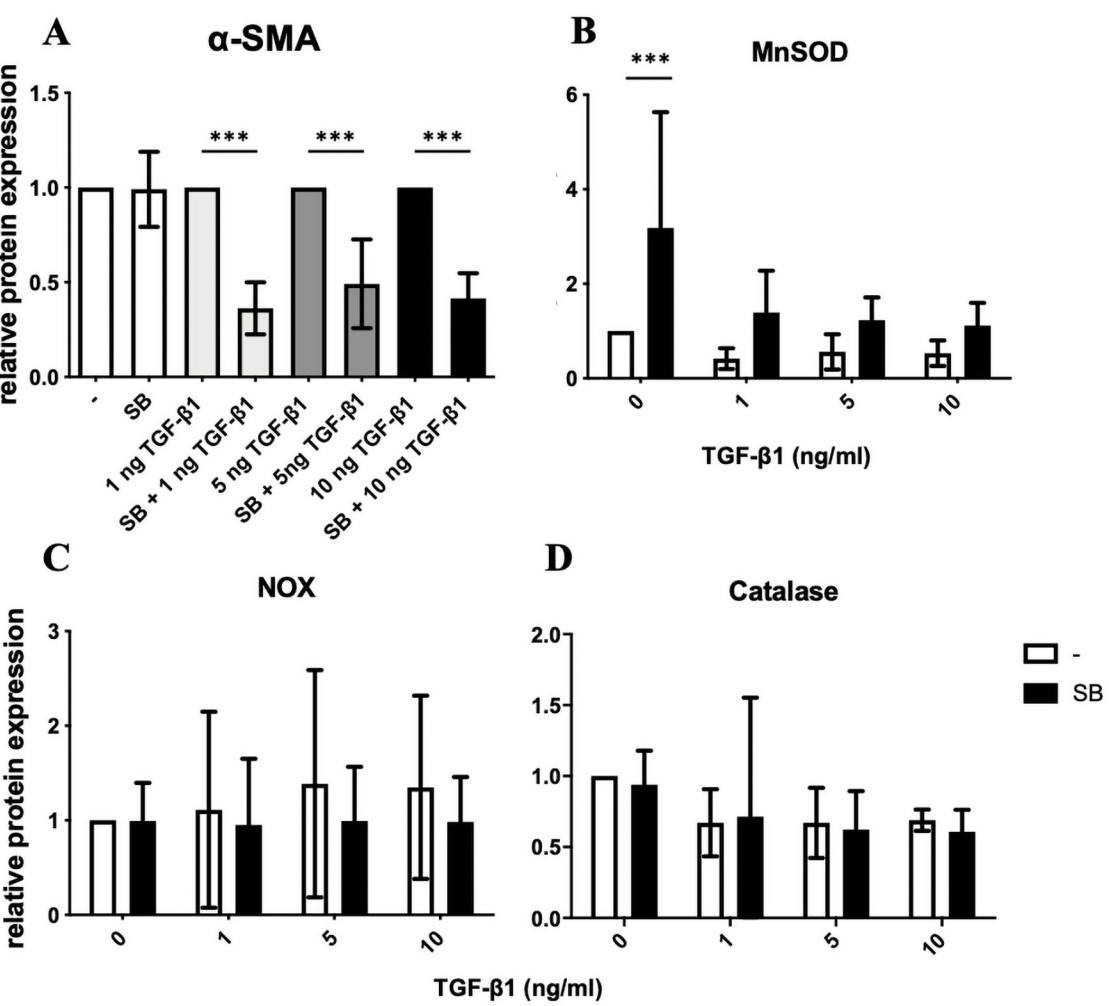

**Fig 6. Relative α-SMA protein expression in SB431542 treated DD fibroblasts (A).** The selective TGF-β1 inhibitor SB431542 reduced α-SMA protein expression significantly, regardless of the TGF-β1 concentrations (A). **Relative MnSOD, NOX4 and catalase protein expression SB431542 treated DD fibroblasts (B–D).** The protein expression of MnSOD was significantly up-regulated by SB 431542 in DD fibroblasts (B). SB431542 had no effect on catalase and NOX4 (C and D). *** $p \leq 0.0005$. Bars represent mean ± SD of individual experiments indicated (n = 5).

because an enhanced ROS burden was already detected in DD compared to CTS fibroblasts. Moreover, this phenomenon was described as one cause for excessive myofibrogenesis in DD [24, 25].

As one major enzyme in the antioxidant defense system, the GPX activity was analyzed [26]. GPX is involved in balancing $H_2O_2$ homeostasis in several signaling pathways (e. g. insulin pathway) and has a dual role in carcinogenesis [27, 28]. Our results show that a blue light irradiation did not affect GPX activity (Fig 2A and 2B).

SOD1 binds copper and zinc ions and catalyzes the conversion of $O_2^-$ into $H_2O_2$ and $O_2$ in mitochondria and cytoplasm. As such an antioxidant isoenzyme plays an essential pathogenic role in several inflammatory diseases [29]. However, the expression of SOD1 is not modified in TGF-β1 activated DD fibroblasts after a blue light treatment neither on day 3, nor on day 5 (Fig 2C and 2D). Riedl et al. evaluated the positive efficacy of a topical gel containing liposomal

encapsulated recombinant human SOD in the treatment of Peyronie's Disease, a painful fibro-proliferative disorder of the penis which is pathophysiologically comparable to DD [30].

The NOX family NADPH oxidases are proteins that share the capacity to transport electrons across plasma and mitochondrial membranes [31, 32]. The biological function of NOX enzymes is therefore the generation of $O_2^-$ and another downstream of ROS ($H_2O_2$). It was documented in human idiopathic fibrosis, that NOX4 derived ROS facilitate TGF-β mediated fibrosis by inducing differentiation of fibroblasts into myofibroblasts [33]. In TGF-β1-activated DD fibroblasts, the NOX4 expression was hardly affected and the additional irradiation only led to a slight increase of NOX4 expression, which was not significant (Fig 2E and 2F). So, it can be assumed that in Dupuytren's myofibrogenesis NOX4 expression had no crucial function.

Catalase promotes the conversion from $H_2O_2$ into $H_2O$ and is able to reduce intracellular ROS level. In our study, it could be demonstrated that, if myofibrogenesis was induced with TGF-β, catalase protein expression was significantly reduced on day 3, (Fig 3A), thereby most likely elevating ROS content. Thus, myofibrogenesis in DD seems to be also mediated by a diminished catalase expression (Figs 2E and 3A). These findings pronounce the thesis, that ROS mediate TGF-β-induced fibrotic disorders [16].

Interestingly, the beneficial blue light irradiation, which inhibited myofibrogenesis evidenced by a decreased α-SMA expression (Fig 1A and 1B), also leads to significant inhibition of catalase in DD fibroblasts (Fig 3A and 3B). In consequence, blue light should increase ROS-level further. In order to prove if increasing ROS level could be responsible for the inhibition of myofibrogenesis, we used increasing non-toxic concentrations of $H_2O_2$ and TGF-β1. The results demonstrated a significantly inhibited α-SMA expression in DD fibroblasts (Fig 4A–4C). This presumption is underlined by our previously published work, demonstrating that blue light irradiation increased ROS content in DD [20]. We assume that the beneficial effect of the blue light irradiation was mediated by an additional heightened ROS level.

Most ROS in cells are generated by the mitochondrial respiratory chain. MnSOD is the major enzyme for modulating mitochondrial ROS homeostasis [34, 35]. Due to its cytoprotective effects, MnSOD degrades the highly reactive and short-half-life $O_2^-$ into the more stable oxidant $H_2O_2$ and $O_2$. Its conventional role of defense against oxidative damage was recently modified by detecting low level MnSOD expression in many human tumors and its role in cancer progression [36]. Our results also showed a significant decreased expression of MnSOD in resting (Fig 5A), in irradiated (Fig 5B), and in irradiated and activated DD fibroblasts compared to CTS fibroblasts. It has already been supposed that mitochondrial dysfunction is involved in DD. Bayat et al. suggest that a mutation in the mitochondrial genome may be predisposing to formation of DD [37]. This suggestion can be supported by the simultaneous presence of mitochondrial mutations in Diabetes mellitus and epilepsy, as both have been associated with a higher risk for DD [38, 39]. In line with this, it has been shown that TGF-β1 increases mitochondrial ROS production in different cell types, which mediate fibrosis related gene expression and myofibroblast differentiation [40]. Moreover, in 1987 the free radical theory for the pathogenesis of DD has been established [10]. Examinations of the palmar fascia of DD, also compared to CTS tissue, showed a sixfold increased hypoxanthine concentration, highest in nodular areas. Furthermore, a higher concentration of XO activity was detected. XO catalyzes the conversion of hypoxanthine to xanthine as well as the conversion of xanthine to uric acid. Both reactions produce short lived $O_2^-$. Murrel et al. substantiate their findings by demonstrating a positive influence of allopurinol, a XO inhibitor in DD patients [41, 42]. Here, we could demonstrate that beside hypoxanthine and XO, also MnSOD seems to be involved in the pathogenesis of DD. This assumption was underlined by our following result: If DD fibroblasts were treated with SB431542, a selective pharmacological inhibitor of

TGF-β-signaling, myofibrogenesis was significantly inhibited (Fig 6A) [43]. But this finding was paralleled by a significantly increased MnSOD protein expression (Fig 6B), thereby most probably elevating long-lasting ROS-level. Apart from that, SB431542 has no effect on catalase and NOX4 (Fig 6C and 6D).

Finally, to elucidate the blue light induced inhibition of myofibrogenesis and pathogenesis of DD, we examined in an additional step the transcription factors, NFκB and ß-catenin (S2 Fig). NFκB is redox-sensitive and has been implicated in cellular response to oxidative stress [44, 45]. In line with this, the blue light irradiation in tendency increased NFκB protein expression, and this effect was significant in DD fibroblasts on day 5 (S1A–S1C Fig).

## Conclusions

Our study illuminates catalase as one key enzyme mediating myofibrogenesis in DD, by increasing ROS level and reducing antioxidant capacity. Also, increasing $H_2O_2$ concentrations prevent myofibrogenesis in DD fibroblasts, as demonstrated by this study.

In summary, myofibrogenesis in DD can be inhibited by blue light irradiation, most probably mediated by increasing intracellular ROS level. Blue light irradiation inhibits the differentiation of fibroblasts into myofibroblasts and may therefore represent a suitable, non-invasive and safe approach in the treatment of DD in future times. However, subsequent studies are required.

## Supporting information

**S1 Fig. CellTiter-blue cell viability assay from Promega to determine non-toxicity of TGF-β1 (1 ng–10 ng/ml) and $H_2O_2$.**
(XLSX)

**S2 Fig. Relative NFκB protein expression (A–C).** In irradiated and activated CTS fibroblasts NFκB protein expression was significantly inhibited in comparison to irradiated CTS fibroblasts on day 3 (B). Moreover, the irradiation elevated NFκB protein expression compared to resting fibroblasts, and this effect was significant in DD fibroblasts on day 5 (C). * $p \leq 0.05$. Bars represent mean ± SD of individual experiments indicated (n = 7). **Relative β-catenin protein expression (D–F).** In irradiated and activated CTS fibroblasts, the β-catenin expression was significantly reduced compared to a solely blue light irradiation (E) on day 3. On day 5, β-catenin expression was significantly increased in activated as well as in irradiated and activated DD fibroblasts (F) compared to resting fibroblasts. * $p \leq 0.05$. Bars represent mean ± SD of individual experiments indicated (n = 8).
(TIF)

## Acknowledgments

Data were provided from the doctoral thesis by Marie Wohltmann, performed in the Department of Orthopedics and Trauma Surgery in the Faculty of Medicine of the Heinrich Heine University Düsseldorf, Germany. We thank Jutta Schneider, Christa-Maria Wilkens, and Samira Seghrouchni for technical assistance. We also thank Prof. Dr. Matthias Born and Dr. rer. nat. Jörg Liebmann (Philips, Innovative Technologies, Aachen, Germany) for transferring us the blue light-emitting device used in our experiments.

## Author Contributions

**Conceptualization:** Carina Jaekel, Lisa Oezel.

**Data curation:** Carina Jaekel, Marie H. Wohltmann, Julia Wille.

**Formal analysis:** Lisa Oezel, Vera Grotheer.

**Investigation:** Vera Grotheer.

**Methodology:** Marie H. Wohltmann, Julia Wille.

**Project administration:** Vera Grotheer.

**Software:** Carina Jaekel.

**Supervision:** Simon Thelen, Vera Grotheer.

**Validation:** Simon Thelen, Joachim Windolf.

**Writing – original draft:** Carina Jaekel, Lisa Oezel.

**Writing – review & editing:** Simon Thelen, Joachim Windolf, Vera Grotheer.

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
