## [Decision Letter · Decision Letter 0]

4 Mar 2021

PONE-D-20-39881

Illuminating the effect of beneficial blue light and ROS-modulating enzymes in Dupuytren’s Disease

PLOS ONE

Dear Dr. Thelen,

Thank you for submitting your manuscript to PLOS ONE. After careful consideration, we feel that it has merit but does not fully meet PLOS ONE’s publication criteria as it currently stands. Therefore, we invite you to submit a revised version of the manuscript that addresses the points raised during the review process.

We look forward to receiving your revised manuscript.

Kind regards,

Michael R Hamblin

Academic Editor

PLOS ONE

Journal Requirements:

2. Please amend either the abstract on the online submission form (via Edit Submission) or the abstract in the manuscript so that they are identical.

3. Please include a copy of Table 1 which you refer to in your text on page 7.

Reviewers' comments:

Reviewer's Responses to Questions

**Comments to the Author**

1. Is the manuscript technically sound, and do the data support the conclusions?

Reviewer #1: Yes

Reviewer #2: Partly

2. Has the statistical analysis been performed appropriately and rigorously? 

Reviewer #1: Yes

Reviewer #2: Yes

3. Have the authors made all data underlying the findings in their manuscript fully available?

Reviewer #1: Yes

Reviewer #2: No

4. Is the manuscript presented in an intelligible fashion and written in standard English?

Reviewer #1: Yes

Reviewer #2: No

5. Review Comments to the Author

Reviewer #1: It is clear that a substantial amount of work was done in this paper, and the topic of using blue light is novel. However, there is still some work to be done before its translatable potential can be assessed. Please see my comments and suggestions:

1. Was any data collected at daily intervals and beyond day 5? It would be interesting to see if there is evidence of a dose-response, and whether it is comparable to the bi-phasic response observed when using other wavelengths.

2. Were there any observable structural changes in vimentin following PBM treatment, and were other cytoskeletal markers investigated? I would expect the data to perhaps correlate with some visual changes in the cytoskeleton such as in actin filaments. It would be interesting to show IF images of fibroblasts throughout the duration of the treatments.

3. It would be useful to have some comments on possible future work from study results, and potential clinical implications this may have as an alternative therapy for the management of Dupuytren's disease.

4. There were some instances of grammatical inconsistencies including informal language used. There also appears to be some factual inconsistencies (e.g. line 84 - a wavelength of 420 nm is MORE not LESS energetic than the 453 nm used in this study) that will need some clarification.

Reviewer #2: This study reports on an interesting topic that would add be a valuable addition to the literature regarding the efficacy of blue light for potential use in a common musculoskeletal condition.

Overall, errors of grammar / punctuation / syntax (and isolated examples of spelling) need to be reviewed and corrected throughout because it detracts from comprehending the justification for the research as raised on the Introduction, and for describing the method and discussing the outcomes in numerous other parts of the paper.

Abstract Line 30: Insert the model in which you evaluated “…the impact of blue light irradiation in human DD fibroblasts…”

Method:

Please clarify how long on average, it took for fibroblasts to reach confluence, and how long between surgical extraction of tissue samples and blue light irradiation.

P6, Lines 138/139: “To evaluate the influence of blue light application, DD and CTS fibroblasts were treated for 3 and 5 days.” Please clarify – does this mean that DD fibroblasts were treated for 3 days and CTS fibroblasts for 5 days? What is the meaning of this stated difference? If the statement means that cells were assessed at day 3 and again at Day 5, provide an explanation for doing so.

P6, Line 146: How was non-toxicity of TGF-β1 and H2O2 established?

P7, line 152: Correct the following: “….supernatant was collected. 25 μl, respectively 12.5 µl supernatant were given…”

IN addition to above point, please carefully check all of the details of the experimental methods to ensure that these are reported correctly. For example, why does “(cf. page 5)” appear at top of page 8? And on line 178, it indicates “fibroblasts were sown and treated for 5 days” which does not agree with P6 Lines 138/139. I note repetition and incorrect grammar also throughout these sections.

P9, Line 194. You state, “TGF-β1 increased α-SMA protein expression on day 3 and 5 in DD fibroblasts.” Does this statement refer to non-irradiated cells? This comment also refers to the caption in Figure 1.

Results:

Throughout the Results section, where a statement is made about significant changes, please provide the p value. For example, p12, line 276: “…protein expression was significantly inhibited (p=XXX)…”

Discussion:

The discussion of the results is confusing and needs to be re-visited. In particular, the authors should restrict their observations only to the research outcomes and avoid hypothesising about unrelated factors.

Conclusion:

New information should not be introduced in the conclusion.

OVERALL: In its present form, this paper cannot be accepted, as a coherent description of the results and their meaning is not provided.

6. PLOS authors have the option to publish the peer review history of their article (what does this mean?). If published, this will include your full peer review and any attached files.

Reviewer #1: **Yes: **Dr Ann Liebert

Reviewer #2: No

---

## [Author Response · Author response to Decision Letter 0]

7 Apr 2021

Response to Reviewers

Journal Requirements:

Answer: Thank you very much for the further instructions. We have revised the style requirements and adjusted accordingly. 

2. Please amend either the abstract on the online submission form (via Edit Submission) or the abstract in the manuscript so that they are identical.

Answer: Sorry for the confusion. We thought entering the abstract is required in both places. In the revision submitted, we will only enter the abstract on the online submission form. 

3. Please include a copy of Table 1 which you refer to in your text on page 7.

Answer: Sorry for forgetting to add the table. We will now include a copy of table 1.

Answer: Corresponding to the Supporting Information guideline we include captions for our Supporting Information files at the end of our manuscript.

Review Comments to the Author:

Reviewer #1: It is clear that a substantial amount of work was done in this paper, and the topic of using blue light is novel. However, there is still some work to be done before its translatable potential can be assessed. Please see my comments and suggestions:

1. Was any data collected at daily intervals and beyond day 5? It would be interesting to see if there is evidence of a dose-response, and whether it is comparable to the bi-phasic response observed when using other wavelengths.

Answer: In the present study, the data were not collected on a daily basis. We collected routinely data on day 3 and 5, because after 5 days the β-TGF1-induced 𝛼-SMA-expression showed nearly maximum expression and the daily radiation over a period of 3 or 5 days was long enough to perform the inhibition of 𝛼-SMA. This effect was the strongest at day 5. In older studies our working group used further wavelengths as 410 nm, 420 nm, 453 nm and 480 nm in dermal fibroblasts, but the wavelength of 410 nm and 420 nm led to toxic effect in a dose and wavelength dependent manner [1]. Furthermore, we compared non-toxic wavelengths as 453 nm and 474 nm and decided to use the more powerful radiation at 453 nm to trigger selective photochemical reactions. Our working group used this wavelength in several experimental setups, for e. g. with Dupuytren fibroblasts published in Plos one evaluating day 3 and day 5, too [2] or with dermal fibroblasts published elsewhere [3]. And for the treatment of Dupuytren fibroblasts we tested different irradiation doses and observed also a dose dependent effect, but according to our own published data, we used a non-toxic dose of 40 Joule/cm2 according to a cell viability assay [2].

In the following we included a picture to make the irradiation and dose dependent effect more visible.

Immunhistochemical analysis of 𝛼-sma-expression (red) in dermal fibroblasts irradiated daily with 453 nm (left) and 474 nm (right) and treated with TGFβ-1 for five days with ascending doses.

1. Opländer C, Hidding S, Werners FB, Born M, Pallua N, Suschek CV. Effects of blue light irradiation on human dermal fibroblasts. J Photochem Photobiol B. 2011 May 3;103(2):118-25. doi: 10.1016/j.jphotobiol.2011.02.018. Epub 2011 Feb 26. PMID: 21421326.

2. Krassovka J, Borgschulze A, Sahlender B, Lögters T, Windolf J, Grotheer V. Blue light irradiation and its beneficial effect on Dupuytren's fibroblasts. PLoS One. 2019 Jan 11;14(1):e0209833. doi: 10.1371/journal.pone.0209833. PMID: 30633751; PMCID: PMC6329497.

3. Taflinski L, Demir E, Kauczok J, Fuchs PC, Born M, Suschek CV, Opländer C. Blue light inhibits transforming growth factor-β1-induced myofibroblast differentiation of human dermal fibroblasts. Exp Dermatol. 2014 Apr;23(4):240-6. doi: 10.1111/exd.12353. PMID: 24533842.

2. Were there any observable structural changes in vimentin following PBM treatment, and were other cytoskeletal markers investigated? I would expect the data to perhaps correlate with some visual changes in the cytoskeleton such as in actin filaments. It would be interesting to show IF images of fibroblasts throughout the duration of the treatments.

Answer: Thank you very much for this advanced idea. So far we have not looked at structural changes or biochemical markers of the cytoskeleton in our work. We would like to raise this aspect in future studies.

3. It would be useful to have some comments on possible future work from study results, and potential clinical implications this may have as an alternative therapy for the management of Dupuytren's disease.

Answer: In der conclusion, we now give a brief outlook on possible future work from our study results. Ad this moment clinical implication of blue light may be a little bit too far away. Further studies should be encouraged to follow this assumption.

Line 389-395: “So, in conclusion, blue light irradiation may represent a suitable and safe approach in the treatment of DD in future times. Due to the invasiveness of currently existing treatment options and the high rate of recurrence, new and alternative therapies are desirable. Our study demonstrated, that the application of blue light could inhibit the differentiation of fibroblasts into myofibroblasts. Subsequent studies that further pursue this approach are urgently required, in order to bring blue light irradiation as a possible noninvasive treatment option for DD into everyday practice.”

4. There were some instances of grammatical inconsistencies including informal language used. There also appears to be some factual inconsistencies (e. g. line 84 - a wavelength of 420 nm is MORE not LESS energetic than the 453 nm used in this study) that will need some clarification.

Answer: Thank you for the careful correction. The factual inconsistencies in line 84 was adjusted.

Reviewer #2: This study reports on an interesting topic that would add be a valuable addition to the literature regarding the efficacy of blue light for potential use in a common musculoskeletal condition.

1. Overall, errors of grammar / punctuation / syntax (and isolated examples of spelling) need to be reviewed and corrected throughout because it detracts from comprehending the justification for the research as raised on the Introduction, and for describing the method and discussing the outcomes in numerous other parts of the paper.

Answer: Thank you very much. The entire manuscript was newly checked on errors of grammar, punctuation and syntax, to avoid these. All changes were marked in the “Revised Manuscript with track changes.

2. Abstract Line 30: Insert the model in which you evaluated “…the impact of blue light irradiation in human DD fibroblasts…”

Answer: The model in which we evaluated was inserted. 

Line 30 -31: “To evaluate the impact of blue light irradiation in an in-vitro model of human DD fibroblasts, myofibrogenesis was induced with TGF-β1.”

3. Please clarify how long on average, it took for fibroblasts to reach confluence, and how long between surgical extraction of tissue samples and blue light irradiation.

Answer: After surgical extraction dupuytren’s tissue was extracted and cut into small pieces in a petri dish (10 cm). After an average time of 14 days the MD-Fibroblasts are confluent. Then proliferation rate exceeds and after additional 14 days we start with the experiments in cell culture passage 3 usually until cell culture passage 5, in exceptional cases until cell culture passage 8. Requirement for using cell culture passage 8 is that fibroblasts proliferation rate and morphology have not changed. 

4. P6, Lines 138/139: “To evaluate the influence of blue light application, DD and CTS fibroblasts were treated for 3 and 5 days.” Please clarify – does this mean that DD fibroblasts were treated for 3 days and CTS fibroblasts for 5 days? What is the meaning of this stated difference? If the statement means that cells were assessed at day 3 and again at Day 5, provide an explanation for doing so.

Answer: We have adjusted the language of the sentence to remove the confusion. Both the DD and CTS fibroblasts were irradiated and examined for 3 and 5 days. There was no difference in the treatment of fibroblasts.

We checked protein expression on day 3 and 5, because after 5 days the β-TGF1-induced 𝛼-SMA-expression showed nearly maximum expression and the daily radiation over a period of 3 or 5 days was long enough to perform the inhibition of 𝛼-SMA. (see Reviewer 1 # 1)

Line 138-139: After irradiation PBS was replaced by fresh media. To evaluate the influence of blue light application, DD as well as CTS fibroblasts were treated for 3 and 5 days.

5. P6, Line 146: How was non-toxicity of TGF-β1 and H2O2 established?

Answer: We had performed a CellTiter-Blue Cell Viability Assay from Promega in order to elucidate, if the combination of TGF-β1 and H2O2 inhibited cell viability. And according to the following result, we conclude that the use of TGF-β1 (1 ng/ml) and H2O2 (10µM, 20, µM, 50 µM) over the course of 5 days was not toxic.

6. P7, line 152: Correct the following: “….supernatant was collected. 25 μl, respectively 12.5 µl supernatant were given…”

Answer: We followed the requirements of the manufacturers protocol, now we mentioned it correctly in the manuscript “According to the manufacturers protocol 25 μl, respectively 12.5 µl supernatant were given in a 96-well plate together with 25 μl, respectively 37.5 µl Assay Buffer

7. IN addition to above point, please carefully check all of the details of the experimental methods to ensure that these are reported correctly. For example, why does “(cf. page 5)” appear at top of page 8? And on line 178, it indicates “fibroblasts were sown and treated for 5 days” which does not agree with P6 Lines 138/139. I note repetition and incorrect grammar also throughout these sections.

Answer: The experiments executed with SB431542 were only performed after 5 days of treatment, because the scientific issue had changed a little. In this case we would like to evaluate the time point which has the highest TGF-β1-induced α-SMA-expression analysed in our work.

8. P9, Line 194. You state, “TGF-β1 increased α-SMA protein expression on day 3 and 5 in DD fibroblasts.” Does this statement refer to non-irradiated cells? This comment also refers to the caption in Figure 1.

Answer: Yes, exactly. This statement refer to non-irritated cells in line to the results in Figure 1. To avoid confusion, we added “non- irritated”. Figure 1 shows: 1.) Untreated fibroblasts, 2.) TGF-β1 stimulated fibroblasts, 3.) Irritated fibroblasts and 4.) TGF-β1 stimulated and irritated fibroblasts.

Line 195: “TGF-β1 increased α-SMA protein expression on day 3 and 5 in non-irradiated DD fibroblasts.”

9. Throughout the Results section, where a statement is made about significant changes, please provide the p value. For example, p12, line 276: “…protein expression was significantly inhibited (p=XXX)…”

Answer: Throughout the entire results section p values were made available for each significant result. Correspondingly inserted passages were highlighted in the text.

10. The discussion of the results is confusing and needs to be re-visited. In particular, the authors should restrict their observations only to the research outcomes and avoid hypothesising about unrelated factors.

Answer: Your proposal has been adopted. The discussion has been revised and adjusted accordingly. Hypotheses relating to factors outside the scope of the present study have been deleted. We hope you find the discussion now shorter and crisper, too. 

11. New information should not be introduced in the conclusion.

Answer: Very valid point. We re-visited the Conclusion which is now limited to a brief summary of the main results without any new information.

12. OVERALL: In its present form, this paper cannot be accepted, as a coherent description of the results and their meaning is not provided.

Answer: We revised the paper thoroughly following the very much appreciated comments of the two reviewers. Apart from corrections of grammar and language flaws, we put a lot of emphasis on making the description of the results and their discussion more coherent and conclusive. We are convinced this added value to this manuscript and we hope it finds your approval.

---

## [Decision Letter · Decision Letter 1]

7 May 2021

PONE-D-20-39881R1

Illuminating the effect of beneficial blue light and ROS-modulating enzymes in Dupuytren’s Disease

PLOS ONE

Dear Dr. Thelen,

Thank you for submitting your manuscript to PLOS ONE. After careful consideration, we feel that it has merit but does not fully meet PLOS ONE’s publication criteria as it currently stands. Therefore, we invite you to submit a revised version of the manuscript that addresses the points raised during the review process.

Reviewer 2 feels that most of the comments were not satisfactorily addressed. Therefore you will be given one more chance to answer these in full. Please take all the comments seriously.

We look forward to receiving your revised manuscript.

Kind regards,

Michael R Hamblin

Academic Editor

PLOS ONE

Reviewers' comments:

Reviewer's Responses to Questions

**Comments to the Author**

1. If the authors have adequately addressed your comments raised in a previous round of review and you feel that this manuscript is now acceptable for publication, you may indicate that here to bypass the “Comments to the Author” section, enter your conflict of interest statement in the “Confidential to Editor” section, and submit your "Accept" recommendation.

Reviewer #1: All comments have been addressed

Reviewer #2: (No Response)

2. Is the manuscript technically sound, and do the data support the conclusions?

Reviewer #1: Yes

Reviewer #2: Yes

3. Has the statistical analysis been performed appropriately and rigorously? 

Reviewer #1: Yes

Reviewer #2: I Don't Know

4. Have the authors made all data underlying the findings in their manuscript fully available?

Reviewer #1: Yes

Reviewer #2: No

5. Is the manuscript presented in an intelligible fashion and written in standard English?

Reviewer #1: Yes

Reviewer #2: No

6. Review Comments to the Author

Reviewer #1: (No Response)

Reviewer #2: I have reviewed the authors’ responses to my comments as well as the highlighted changes in the re-submitted manuscript, and I feel that very little has been changed in their response to my feedback. Specifically:

No information has been added TO THE MANUSCRIPT regarding my comment about “Please clarify how long on average, it took for fibroblasts to reach confluence, and how long between surgical extraction of tissue samples and blue light irradiation”

No information has been added TO THE MANUSCRIPT regarding my comment about “How was non-toxicity of TGF-β1 and H2O2 established?”

Grammar remains an issue with this paper. Some examples follow (but there are many more which need to be identified and corrected), and I believe that a native English speaking editor would be able to assist with these matters to aid clarity:

Page 6: “Prior delivery the irradiance of the LED-device was proven using an integrating (Ulbricht) sphere.” Do you mean, “Prior to use, the irradiance of the LED-device was verified using an integrating (Ulbricht) sphere.”?

Page 7: The following sentence remains unclear: “According to the manufacturer’s protocol 25 μl, respectively 12.5 μl supernatant were given in a 96-well plate together with 25 μl, respectively 37.5 μl Assay Buffer.”

Page 7: “Respectively, 10 μg or rather 20 μg protein were mixed…” So, which was it: 10 μg or 20 μg? Please be accurate.

Page 7: “…anti-Western marker in TBST, who was added for 1 h (RT).” Rather than the use of word “who”, it should probably be the word “that”.

Page 8: I see that the following is still in place: “(cf. page 5)”. I have looked at page 5 and I am unable to discern what this means.

Page 12: Use of the word “erased” is inappropriate both here and in a caption to figure S 1 Fig. Relative NFκB protein expression (A – C). Please replace with a more meaningful word.

Non-scientific language and colloquialisms such as those used on page 13 (as examples) are not appropriate. Please replace:

“remarkable” (do you mean that the assumption was prescient, or something to be remarked upon?)

Starting a sentence with “Anyway” is inappropriate

Page 13: “As such an antioxidant isoenzyme, it plays an…” (Delete “it”)

Page 13: The sentence beginning with “Whereas Riedl et al. evaluated the positive efficacy of a topical gel containing liposomal encapsulated recombinant…” is not complete and needs to be reviewed.

Page 14: The following does not make sense: “In TGF-β1-activated DD fibroblasts, the NOX4 expression was hardly affected and the additional irradiation slightly gained NOX4 expression…” Clarify please.

Page 14: The following phrase is not clear: “…further accomplishes leads to significant inhibition of…” Please clarify

Page 14, Line 332: Replace ‘mayor” with “major”

Page 15, Line 341: Replace “und” with “and”

Page 15, final sentence: Describing something as “rather significant” is not appropriate in a scientific sense. It is either significant or not.

Further, one matter that was not obvious to me the first time around:

- the temperature of cell culture plates in the experimental groups “never exceeded 38oC” suggests that there was potentially a marked increase in temperature above room temperature that could have affected the results. It also suggests that there was a range of different temperatures across the experimental conditions which may theoretically have affected the results. Firstly, please add to the manuscript what the temperature range was and the mean temperature for each experimental condition. The large differential in temperature with the control plates (kept at 25oC) suggests that this could have been a confounder to the results. Can the authors explain IN THE MANUSCRIPT whether this apparent temperature differential may have had an effect on the outcomes?

7. PLOS authors have the option to publish the peer review history of their article (what does this mean?). If published, this will include your full peer review and any attached files.

Reviewer #1: **Yes: **Dr Ann Liebert

Reviewer #2: No

---

## [Decision Letter · Decision Letter 2]

14 Jun 2021

Illuminating the effect of beneficial blue light and ROS-modulating enzymes in Dupuytren’s Disease

PONE-D-20-39881R2

Dear Dr. Thelen,

We’re pleased to inform you that your manuscript has been judged scientifically suitable for publication and will be formally accepted for publication once it meets all outstanding technical requirements.

Kind regards,

Michael R Hamblin

Academic Editor

PLOS ONE

Additional Editor Comments (optional):

Reviewers' comments:

Reviewer's Responses to Questions

**Comments to the Author**

1. If the authors have adequately addressed your comments raised in a previous round of review and you feel that this manuscript is now acceptable for publication, you may indicate that here to bypass the “Comments to the Author” section, enter your conflict of interest statement in the “Confidential to Editor” section, and submit your "Accept" recommendation.

Reviewer #2: All comments have been addressed

2. Is the manuscript technically sound, and do the data support the conclusions?

Reviewer #2: Yes

3. Has the statistical analysis been performed appropriately and rigorously? 

Reviewer #2: Yes

4. Have the authors made all data underlying the findings in their manuscript fully available?

Reviewer #2: Yes

5. Is the manuscript presented in an intelligible fashion and written in standard English?

Reviewer #2: Yes

6. Review Comments to the Author

Reviewer #2: Thank you to the authors for addressing the points made in the earlier review process.

Regarding point 18. I wrote: "Further, one matter that was not obvious to me the first time around: - the temperature of

cell culture plates in the experimental groups “never exceeded 38°C” suggests that there was potentially a marked increase in temperature above room temperature that could have affected the results. It also suggests that there was a range of different temperatures across the experimental conditions which may theoretically have affected the results. Firstly, please add to the manuscript what the temperature range was and the mean temperature for each experimental condition. The large differential in temperature with the control plates (kept at 25oC) suggests that this could have been a confounder to the results. Can the authors explain IN THE MANUSCRIPT whether this apparent temperature differential may have had an effect on the outcomes?"

You answered: "We are sorry for causing a misunderstanding here. It was not the case that the irradiated plates were incubated at 38 °C and the control plate at 25 °C for an extended period of time. The experimental procedure was conducted in the following way: The cell culture plates were in general incubated in the humidified atmosphere at 37 °C. For the short irradiation time (16.7 min) irradiated CTS and DD fibroblasts were exposed to room temperature (18 – 21 °C) under the LED device and for the same duration of time control DD and CTS fibroblasts were stored in a Biometra OV3 Hybridisation oven at 25°C, in order to have comparable temperatures in both experimental setups (+/- irradiation). This was evaluated in control experiments using a digital thermometer, determining that irradiated DD and CTS fibroblasts never exceeded a temperature of 38 °C and control fibroblasts were kept at a temperature of 37 °C. We specified this procedure in the manuscript. This does not mean that the range of temperature was varying so much. Maybe we better erase the last sentence, if it leads to confusion?"

Firstly, than you for clarifying and being accurate. Please don't change the wording.

Secondly: You should know that the method you used may be criticized by readers due to the temperature differences outlined in your methods. In the field of photobiomodulation therapy, temperature change (rather than the effect of the light per se) is often seen as the reason why differences in effects may occur. Your future work in this space should seek to ensure that the temperatures at which the control and experimental conditions are handled are as close as possible. You may wish to acknowledge this factor in your Discussion section.

7. PLOS authors have the option to publish the peer review history of their article (what does this mean?). If published, this will include your full peer review and any attached files.

Reviewer #2: **Yes: **E-L. Laakso

---

## [Editor Report · Acceptance letter]

24 Jun 2021

PONE-D-20-39881R2 

Illuminating the effect of beneficial blue light and ROS-modulating enzymes in dupuytren’s disease 

Dear Dr. Thelen:

I'm pleased to inform you that your manuscript has been deemed suitable for publication in PLOS ONE. Congratulations! Your manuscript is now with our production department. 

Kind regards, 

on behalf of

Dr. Michael R Hamblin 

Academic Editor

PLOS ONE